# Identification of Weed-Suppressive Tomato Cultivars for Weed Management

**DOI:** 10.3390/plants11030411

**Published:** 2022-02-02

**Authors:** Isabel Schlegel Werle, Edicarlos Castro, Carolina Pucci, Bhawna Soni Chakraborty, Shaun Broderick, Te Ming Tseng

**Affiliations:** 1Department of Plant and Soil Sciences, Mississippi State University, Starkville, MS 39762, USA; iswerle@uark.edu (I.S.W.); ebd88@msstate.edu (E.C.); carolpuccim@gmail.com (C.P.); bhawnasoni23@gmail.com (B.S.C.); 2Truck Crops Experiment Station, Mississippi State University, Crystal Springs, MS 39059, USA; srb559@msstate.edu

**Keywords:** crop improvement, large crabgrass (*Digitaria sanguinalis* L.), Palmer amaranth (*Amaranthus palmeri* S. Wats), yellow nutsedge (*Cyperus esculentus* L.), weed suppression

## Abstract

Weed-suppressive crop cultivars are a potentially attractive option in weed management strategies (IWM). A greenhouse study was conducted at the R. R. Foil Plant Science Research Center, Starkville, MS, to assess the potential weed-suppressive ability of 17 tomato cultivars against Palmer amaranth (*Amaranthus palmeri* S. Wats), yellow nutsedge (*Cyperus esculentus* L.), and large crabgrass (*Digitaria sanguinalis* L.). The experiment was a completely randomized design, with four replications, and was repeated twice. The height, chlorophyll, and dry weight biomass of the weeds were measured 28 days after sowing. Weed suppression varied greatly among tomato cultivars. The most significant effect of tomato interference was recorded on Palmer amaranth, and the least reduction was observed with yellow nutsedge plants. Cultivars 15 and 41 reduced Palmer amaranth height and biomass by about 45 and 80%, respectively, while cultivar 38 reduced 60% of the chlorophyll percentage. Large crabgrass plants were 35% shorter in the presence of cultivar 38 and had a biomass reduction of 35% in the presence of cultivar 38. Under tomato interference, a minimal effect was observed in chlorophyll, height, and biomass of yellow nutsedge seedlings. Factoring all parameters evaluated, cultivars 38 and 33 were most suppressive against Palmer amaranth and large crabgrass.

## 1. Introduction

Tomato is an economically important vegetable in the United States. In 2019, there was 110,700 ha of processing tomato harvested, with a national average yield of 44,000 kg ha^−1^ [1]. Weed management is one of the costliest practices in tomato production, and it is considered a significant portion of the total operating cost to farmers [2]. Notably, weed thresholds acceptable to growers of high-value vegetables such as fresh and processed tomato are near zero. 

Palmer amaranth (*Amaranthus palmeri* S. Wats.), yellow nutsedge (*Cyperus esculentus* L.), and large crabgrass (*Digitaria sanguinalis* L.) are among the primary weed species interfering in tomato farming [3]. The season-long presence of 25 yellow nutsedge plants m^2^ can reduce tomato yield by 25% [4]. Furthermore, field infestations of large crabgrass at a density of 55 plants m^2^ in direct-seeded tomato can cause up to 74% yield reduction [5]. In transplanted tomato crops, a decrease of 76% was observed in productivity under infestation of *Amaranthus* spp. [6]. Physical and chemical interactions govern the interference of these weed species. Essentially, plant–plant interaction comprises two factors: allelopathy and competition [7]. Competition is the physical perception of surrounding environmental resources available. At the same time, allelopathy is a chemical-mediated interference associated with the release of compounds from a donor plant that can influence the growth and performance of a receiving plant [8]. Resource competition has driven plant community interactions, but, recently, allelopathy has emerged as an approach to solve issues in agricultural fields. 

Attention has been given to identifying a wide range of crops with natural weed-suppressive ability to offset weed interference. Weed-suppressive ability, within the context of this paper, refers to the potential of a crop to reduce or inhibit weed emergence or growth. Certain tomato cultivars have shown differential weed-suppressive abilities when grown together with weeds compared to monocultures. For instance, tomato cultivars can reduce barnyardgrass seed production, but the magnitude of this reduction depends on barnyardgrass density and tomato sowing rate [9]. Cultivar differences in weed-competitiveness were documented among four tomato cultivars in tomatoes in response to velvetleaf competition, where cultivar H8892 had the lowest yield loss due to weed interference [10]. Under full tomato interference, the shoot dry weight of yellow nutsedge plants was reduced by 48% compared with the weed growing in monoculture. Additionally, the belowground plant parts were affected, where yellow nutsedge plants produced 20% fewer and 40% smaller tubers than when grown in the absence of tomato [4]. Tomato cultivars 9492, 9553, and 9992 are proven to have considerable tolerance to lespedeza dodder (*Cuscuta* spp.), a parasitic weed, resulting in dodder growth reduction by more than 70% [11]. The utility of such cultivars with specific abilities to tolerate weed infestation can be valuable in low-input agricultural systems or situations when chemical weed control is not possible, such as in organic cropping.

Weed-suppressive cultivars involve a manifestation of joint activity and interaction of many characteristics instead of a single trait [12]. Plant architecture, growth habit pattern, and overall morphological performance, such as early groundcover and leaf area accumulation, are essential traits responsible for increased competitiveness between crop and weed species [12]. Weed-suppressive crops are often found to present allelopathic properties. The allelopathic ability has been found in cereal crops, such as rye, sorghum, rice, and wheat, and leguminous crops, such as sunflower and rapeseed [13]. Some well-studied phytochemicals include simple phenolics, flavonoids, and alkaloids [14]. The allelopathic property of some plants is potentially valuable for intercropping systems, soil additives via crop residue incorporation, and suppression of weed emergence [15]. The discovery of parental varieties with weed-suppressive potential can be a helpful resource to tomato breeding programs and benefit farmers with an alternative to chemical weed control, thus contributing to a more sustainable farming system. The present research evaluated seventeen tomato genotypes with potential weed-suppressive ability against problematic weeds in tomato production. The hypothesis underlying tomato was that tomato has the potential to suppress surrounding weeds through a higher competitive ability or allelopathy, against the null hypothesis that the tomato has no potential to suppress weed. This study, therefore, aims (i) to determine the weed-suppressive potential of diverse tomato germplasm, and (ii) to examine the effects of weed–crop interaction of tomato cultivars on Palmer amaranth, junglerice, and yellow nutsedge. 

## 2. Materials and Methods

Greenhouse experiments were conducted over three years (2017 to 2019) at the Mississippi State University at the R. R. Foil Plant Science Research Center (88.7847°, 33.4552°), Starkville, MS. Weed-suppressive potential of 17 tomato cultivars (Table 1) were tested against Palmer amaranth (*Palmer amaranth* S. Wats), yellow nutsedge (*Cyperus esculentus* L.), and large crabgrass (*Digitaria sanguinalis* L.). Weed-suppressive ability was evaluated following the method described by Shrestha et al. (2020) [16], with modifications. 

Pots of 10 L were filled with a mixture of field soil and commercial potting mix (2:1). Field soil was used as the growth medium to minimize chemical inhibition by using the organic substrate. To avoid water contact with plant shoot, the pots were placed in trays filled with water according to their necessity. Four tomato plants and four plants of a single weed species were colocated in the same potting container (Figure 1). Thus, tomato plants could interfere with weed species either by competition or by generating chemical interference due to the release of allelochemicals from root exudates. In our context, this screening involves tomato cultivars that are able to suppress the target weeds due to their robust morphological and natural genetic traits. Tomato and weeds were direct-seeded at equal spacing. Tomato seeds were placed on the edge of the pot, while the seeds of the weed species were sown at the center of the pot. At the moment of emergence, four tomato plants and four weed plants were kept per pot.

The experiment was conducted in a completely randomized design, with four replications, and was repeated twice for each cultivar and weed species. Greenhouse day/night temperature was set at 30/25 °C, and humidity was maintained at 70%. The four central weed plants were considered for the evaluations. Plant height, chlorophyll, and dry weight biomass of the weeds were measured 28 days after sowing (DAS). The height of the weeds was measured from the soil to the insertion of the last leaf. Chlorophyll was evaluated using a CCM-300 SPAD meter (Opti-Sciences Inc., Hudson, NY, USA). At 28 DAS, plants were cut at the soil surface and stored in paper bags. Samples were dried in a forced-air circulation oven at 60 °C until constant weight. Comparison among weed species was based on percent inhibition data. Sixteen plants of each weed species were grown as a control treatment without tomato interference. All variables mentioned above were recorded to calculate the reduction, as shown in the equation below (1). Height, chlorophyll, and biomass reduction percentage of recipient plant and donor plant samples were calculated as
reduction (%)=100 – (receiver plant × 100)÷(control plant)
where the control is the mean height, chlorophyll, or biomass of all the plants in the control plants combined, and the height, chlorophyll, or biomass receiver is based on weed plant grown with tomato. Principal component analysis (PCA) was used to determine the highest- and lowest-ranking tomato cultivars against each weed species based on height, chlorophyll, and biomass. Data were analyzed using a general linear model with mean values separated using Fisher’s protected least significant difference at a 0.05 probability level using JMP 16.0 software (SAS Institute Inc., Cary, NC, USA).

## 3. Result

The weed-suppressive potential of tomato accessions was calculated based on height, shoot dry biomass, and chlorophyll reduction of three weed species. The greenhouse study was conducted over four weeks. Weed height, chlorophyll, and shoot biomass were significantly affected by the interference of tomato cultivars (*p* < 0.05), and the null hypothesis was rejected. Chlorophyll reduction percentage was relatively low for all weed species evaluated (Table 2). The chlorophyll reduction of Palmer amaranth ranged from 10 to 60%, and cultivar 38 caused the highest reduction (*p* = 0.0001) (Table 2). Yellow nutsedge seedlings had less than 20% of chlorophyll reduction (*p* = 0.0001). Cultivars 63 and 5 reduced yellow nutsedge chlorophyll the most, while roughly half of the cultivars tested had less than 10% chlorophyll reduction. None of the tomato cultivars presented a considerable reduction in the chlorophyll of large crabgrass plants (<25%). 

Palmer amaranth was the most affected by tomato cultivars among the weed species (Figure 2). Height reduction of Palmer amaranth (*p* = 0.0001) ranged from 18 to 45%. Cultivars 15 and 41 stunted Palmer amaranth height the most (45 and 44%, respectively), but were statistically similar to the other cultivars (Figure 3). Although no significant differences (*p* = 0.05) were found among tomato cultivars, cultivar 20 stunted yellow nutsedge the most, 77% more than cultivars 5, 44, 59, and 54. Overall, a range of 20 to 35% of height reduction was found in large crabgrass plants. Large crabgrass height was reduced by 35% in the presence of cultivar 38 compared with the weed in monoculture, which did not differ statistically from the other cultivars.

Palmer amaranth shoot biomass was considerably decreased due to the interference of tomato cultivars (*p* = 0.0001), and the percentage of reduction ranged from 25 to 80% (Figure 4). The maximum reduction of dry biomass in Palmer amaranth was due to the influence of the tomato cultivars 33 (83%) and 15 (83%), whereas the minimum biomass reduction was due to cultivar 10 (25%) (Figure 4). Yellow nutsedge biomass decreased about 40% with the interference of cultivar 15, but most of the cultivars did not reduce the biomass by more than 30% (*p* = 0.0048). Overall, 60% of tomato cultivars resulted in biomass reduction of large crabgrass by more than 20% (*p* = 0.0137). The highest biomass reduction in large crabgrass among the cultivars was due to cultivar 63 (45%), followed by cultivar 64, 33, and 38, with about 40% reduction. The least suppressive effect was observed with cultivar 18 (5%). 

A principal component analysis (PCA) was performed to identify the most contributing traits in suppressing weed species accurately. Principal component 1 contributed 55% of the total variability of large crabgrass, whereas 33.2% of the variation can be attributed to component 2 (Figure 5). In Palmer amaranth, principal component 1 (PC1) accounted for 53.5% of the total variation in the dataset, and PC2 accounted for 36.4%. The PCA of yellow nutsedge revealed that 59.4% of the variation in allelopathic potential was related to component 1, and 30.5% was related to component 2. Among the parameters used, height reduction and biomass reduction were positively correlated with component 1, but chlorophyll reduction was not closely related to these parameters. From the PCA analysis, tomato cultivars 38 and 63 clustered together in the PC1, indicating high weed-suppressive potential on large crabgrass plants. Cultivars 38, 59, and 33 exhibited high suppression on Palmer amaranth. Yellow nutsedge was affected the most by cultivars 7, 10, and 17.

## 4. Discussion

Plants respond differently to stressful conditions of interference by neighboring species [17,18]. The receiver plant suppression in our experimental design can occur partly due to competition and allelopathic interferences. Our study is only a first step towards a more comprehensive analysis of crop–weed interaction. At this research stage, we only considered the aboveground parts of the plant, not including root system or soil environmental parameters. Nevertheless, this approach explores the processes in plant–plant interactions, allowing for the analysis of different architectural and morphological interactions among tomato genotypes and distinct weed species.

The competitive ability of a plant is associated with the space that it is able to occupy at the early season stage and the rate that this plant is able to expand within this space under limiting resources [19]. Because growth morphology differs among weed species, junglerice, yellow nutsedge, and Palmer amaranth were chosen in this study as representatives of Poaceae and Amaranthaceae families. In tomato, the critical period of weed interference was found to occur between 28 and 35 days after transplanting [20], 28 to 45 days [6], or 24 to 36 days after transplanting [21]. Tomatoes are typically not directly sown in the soil to provide an early-season advantage. However, for this experiment, tomato and weed seeds were sown simultaneous, allowing full interference for 28 days. 

Chlorophyll content was reduced by less than 25%, regardless of weed species and tomato cultivars. In a screening with potential allelopathic varieties of cultivated rice and weedy rice against barnyardgrass (Echinochloa crusgalli), chlorophyll reduction of barnyardgrass plants was little and ranged from 3–34% [22]. Under interference and stress conditions, plants tend to defend themselves and increase competitiveness by allocating resources to energy-costly defense mechanisms [23]. For instance, rather than investing in growth and biomass production, plants tend to present stunting due to stress conditions [24,25]. This phenomenon is widely studied with insect–plant or pathogen–plant interactions [26], but few studies have described plant–plant interference either by allelopathy or competition. Our results show that the interference caused by the tomato cultivars on weed species was entirely dependent on the weed species. Palmer amaranth and large crabgrass were the most affected under the interference of tomato cultivars, while the reduction of yellow nutsedge growth was less than 10% across cultivars. In studies with yellow and purple nutsedge, a higher effect of crop–weed interaction was observed with yellow nutsedge. Overall, tomato shoot biomass was reduced by 19%, and this result was attributed to increased aboveground competition between tomato and yellow nutsedge, while the growth of purple nutsedge had a higher effect by tomato shading [4]. In this study, yellow nutsedge shoots were above the tomato canopy (data not shown), indicating that competition for light between the crop and the weed could contribute to the lower height reduction observed in this species. The low reduction in plant height clearly indicates yellow nutsedge to be the stronger competitor in the early growth stage of tomato plants, which is also supported by the low dry biomass reduction of yellow nutsedge plants compared to Palmer amaranth and junglerice.

Except for cultivar 10, the shoot biomass of Palmer amaranth was decreased by more than 50% regardless of the tomato cultivar. Tomato cultivars are known to provide suppression of Palmer amaranth under full interference. In previous research, Palmer amaranth shoot dry biomass in monoculture decreased from 440 g to 230 g per plant when grown with tomato [27]. Other studies also showed that Palmer amaranth growth was suppressed by tomato cultivars regardless of weed population density, which reinforces the feasibility of weed-suppressive crops as a tool for weed management where this weed is a problem [28]. Our results indicated that large crabgrass and yellow nutsedge exhibited no more than 30% of biomass reduction when grown in the presence of the tomato. With the simultaneous emergence of tomato and barnyardgrass, a greater competition from barnyardgrass is expected [29]. In these circumstances, the competition for light can be a major determinant of reduction in tomato yield due to barnyardgrass interference. Nevertheless, the competition can increase when barnyardgrass plants are clumped compared to when they are either uniformly or randomly distributed [29]. In our study, junglerice plants were uniformly distributed when grown together with tomato plants, which can possibly explain the low effect of tomato on junglerice growth. 

The genetic background of the tomato genotypes can significantly influence the weed–crop interactions. The natural weed-suppressive ability varied within tomato cultivars. The influence of these cultivars also varied among each parameter measured, and not all the cultivars had high weed-inhibition potential. Tomato cultivars 38 and 33 were highly suppressive in this screening based on all the three parameters measured. Although we do not have enough evidence to indicate allelopathy as the phenomenon observed in this study, this enhanced weed-suppressive potential of these two tomato cultivars might be caused by robustness or allelopathic traits. The greatest suppressive potential was observed on Palmer amaranth. In contrast, very little effect was observed on yellow nutsedge seedlings. Overall, these findings are encouraging, as they show that some tomato cultivars are likely to have a significant impact on weed suppression. Improving crop competitiveness is one of the principles behind cultural weed-control management and a valuable step forward for low-input systems and resource-poor farmers. Tomato growers can benefit from this research by selecting genotypes with advantaged characteristics against weeds, especially in fields where Palmer amaranth is a problem.

## 5. Conclusions

This study demonstrated that tomato cultivars can suppress the growth of key weed species in tomato production. Cultivars 38 and 33 exhibited the highest weed suppression based on the three parameters measured. Under interference, weeds showed a reduction in growth and biomass accumulation, but no effects in chlorophyll reduction were observed. Palmer amaranth growth and biomass were greatly influenced due to interference of tomato cultivars, whereas yellow nutsedge plants showed very little effect. Altogether, our study provides evidence that weed-suppressive tomato genotypes can be integrated into weed management programs and should be further studied to understand what mechanisms are most likely associated with the enhanced potential to withstand weed interference. 

## Figures and Tables

**Figure 1 plants-11-00411-f001:**
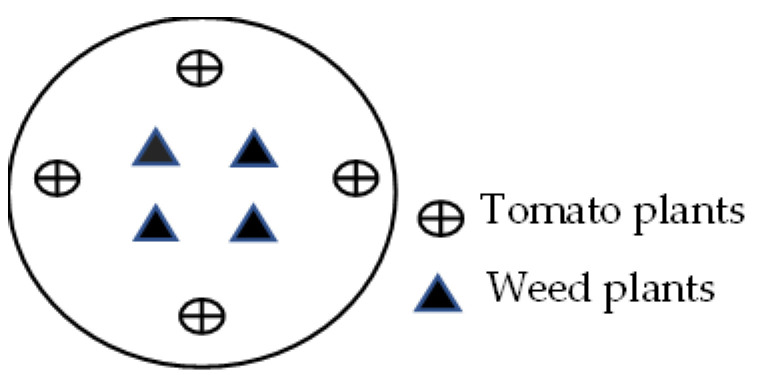
Representation of the experimental setup to evaluate weed-suppressive ability of tomato genotypes against Palmer amaranth, large crabgrass, and yellow nutsedge. Four tomato plants were sown at the edges of the pot, and four plants of a particular weed species were sown in the center of the pot.

**Figure 2 plants-11-00411-f002:**
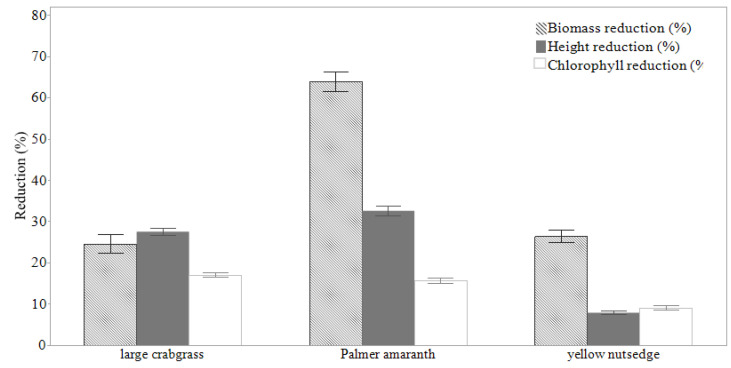
Average seedling height, biomass, and chlorophyll reduction (%) of Palmer amaranth, yellow nutsedge, and large crabgrass, at 28 days after sowing. Four tomato plants and four plants of a single weed species were grown together in one pot. The reduction values were based on a comparison to plants of individual weed species grown as a control treatment without tomato interference. Error bars represent the standard deviation of the mean.

**Figure 3 plants-11-00411-f003:**
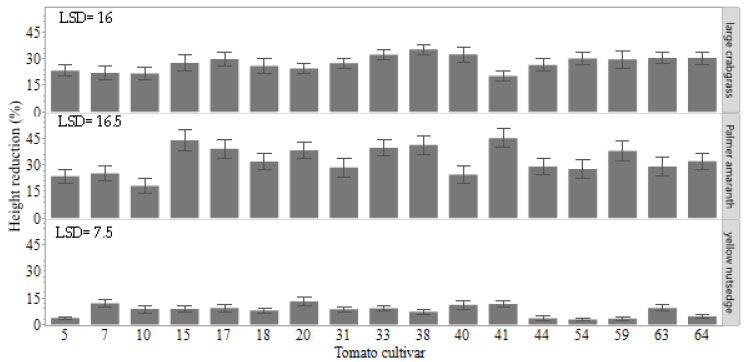
Weed-suppressive ability of tomato cultivars on height reduction (%) and chlorophyll reduction (%) of Palmer amaranth, large crabgrass, and yellow nutsedge at 28 days after sowing. Four tomato plants and four plants of a single weed species were grown together in the same pot. The reduction values were based on a comparison to plants of individual weed species grown as a control treatment without tomato interference. Error bars represent the standard deviation of the mean. The difference between the two mean values is compared to the least significant difference (LSD) value. If the difference was greater than the LSD value, then the means are significantly different according to Student’s *t*-test at a 0.05 probability level.

**Figure 4 plants-11-00411-f004:**
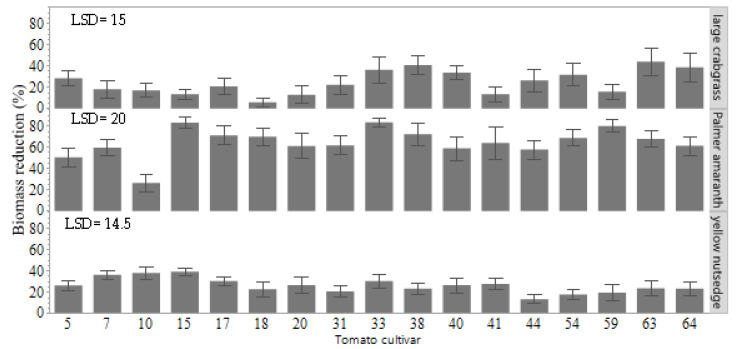
Weed-suppressive ability of tomato cultivars on biomass reduction (%) of Palmer amaranth, large crabgrass, and yellow nutsedge at 28 days after sowing. Four tomato plants and four plants of a single weed species were grown together in the same pot. The reduction values were based on a comparison to plants of individual weed species grown as a control treatment without tomato interference. Error bars represent the standard deviation of the mean. The difference between the two mean values is compared to the least significant difference (LSD) value. If the difference was greater than the LSD value, then the means are significantly different according to Student’s *t*-test at a 0.05 probability level.

**Figure 5 plants-11-00411-f005:**
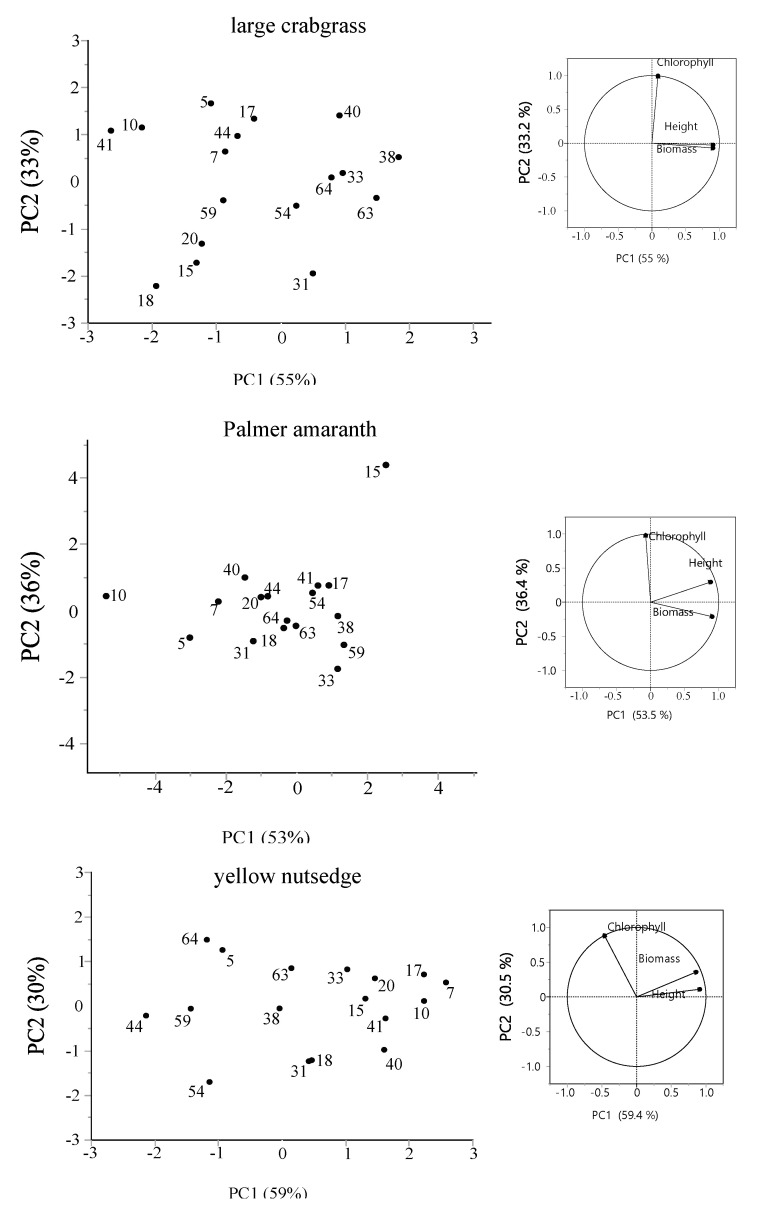
Principal component analysis (PCA) is based on three components: height, chlorophyll, and biomass. The proportion of variances for principal components (PC) 1 and 2 are shown in parentheses.

**Table 1 plants-11-00411-t001:** Codes and names of 17 tomato cultivars tested for suppressive ability on Palmer amaranth, yellow nutsedge, and large crabgrass.

Cultivar Code	Cultivar Name
5	AVTO 9802
7	1595
10	114
15	1511
17	2079
18	2709
20	1512
31	1458
33	2661
38	168
40	3056
41	2401
44	1511
54	M82
59	FERRY MORSE
63	AVTO 1219
64	WV63

**Table 2 plants-11-00411-t002:** Percentage of chlorophyll reduction of Palmer amaranth, yellow nutsedge, and large crabgrass across 17 tomato cultivars, at 28 days after sowing. Four tomato plants and four plants of a single weed species were grown together in the same pot. The reduction values were based on a comparison to plants of individual weed species grown as a control treatment without tomato interference. Error bars represent the standard deviation of the mean. The difference between the two mean values is compared to the least significant difference (LSD) value. If the difference was greater than the LSD value, then the means are significantly different according to Student’s *t*-test at a 0.05 probability level.

	Chlorophyll Reduction (%)		
Cultivar	Palmer Amaranth	Yellow Nutsedge	Large Crabgrass
5	14	17	18
7	16	8	19
10	11	4	16
15	13	11	9
17	17	1	21
18	13	2	9
20	21	15	8
31	16	2	13
33	10	5	17
38	61	10	20
40	11	2	8
41	13	12	11
44	12	5	23
54	16	8	21
59	16	14	18
63	10	17	11
64	13	14	14
LSD (α = 0.05)	7.5	10	8

## Data Availability

Not applicable.

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
