# Peer review of "Identification of Weed-Suppressive Tomato Cultivars for Weed Management"

_plants, 2022, doi:10.3390/plants11030411_

Round 1

Reviewer 1 Report

Dear Authors,

This manuscript has been partially revised to consider allelopathic phenomena and competition in weed control. The introduction has been significantly modified to focus on weed suppression rather than allelopathy itself. The methodological part was also modified. However, most of the goals identified were not achieved, including:

  • There is still no clear, clear purpose of the work.
  • The research hypothesis was not specified.
  • The discussion of the research results was only partially modified.
  • The discussion was slightly changed, but not all aspects of the work were discussed.
  • The soil tests before setting up and after decommissioning the experiment were performed to a limited extent. The authors only stated that: "The lighting and water conditions were optimal, and a similar environment was provided for all experimental units to reduce data variability." No details of these conditions were given. Soil and water are essential for the production of allelopathic substances. The soil was not studied in this experiment and no attempts were made to determine these compounds or their effect on plants.
  • The assumptions of the work were poorly thought out because there were too many uncontrolled factors in the experiment and too little data for this experiment to give reliable results.
  • It is premature to publish the work at this stage. More experimental factors should be carefully controlled to eliminate experimental bias.

Author Response

There is still no clear, clear purpose of the work.

A: We have specified the overall objective of the study in the introduction (lines 84-86)

The research hypothesis was not specified.

A: A hypothesis has been added in the material and methods (lines 95-97)

The discussion of the research results was only partially modified. The discussion was slightly changed, but not all aspects of the work were discussed.

A: Significant changes have been made in the discussion section of this paper.

The soil tests before setting up and after decommissioning the experiment were performed to a limited extent. The authors only stated that: "The lighting and water conditions were optimal, and a similar environment was provided for all experimental units to reduce data variability." No details of these conditions were given. Soil and water are essential for the production of allelopathic substances. The soil was not studied in this experiment and no attempts were made to determine these compounds or their effect on plants. The assumptions of the work were poorly thought out because there were too many uncontrolled factors in the experiment and too little data for this experiment to give reliable results. It is premature to publish the work at this stage. More experimental factors should be carefully controlled to eliminate experimental bias.

A: We have included information about the temperature and humidity conditions in the material and methods. Although no attempt was made to investigate the soil effect in the study, we have tried to make it clear to the readers that this is a study considering only the aboveground plant parts of the weed and that there is no certainty whether the results are attributed to allelopathic properties, competition ability, or both. It is our understanding that further investigation is needed to clarify these aspects.

Reviewer 2 Report

The authors improved the manuscript.

Some issues are still not addressed in this work:

  1. We know that the volume of pots was 10 L, but what was the area of the pots with tomato and weeds?
  2. What was the emergence time of tomatoes and weeds, as this could affect their competition significantly?
  3. The authors underline, that the results of this study could be of importance for tomato growers, but in which countries? Maybe only in the US or even a specific state. I suggest adding this information in the abstract.

Author Response

Some issues are still not addressed in this work:

We know that the volume of pots was 10 L, but what was the area of the pots with tomatoes and weeds?
A: The diameter of the pot was 10 3/8 inches or an area of 95 square inches. Since manuscripts usually indicate the pot volume, we did not include the pot dimensions. We can, however, include the dimensions if the editor/reviewer prefers.

What was the emergence time of tomatoes and weeds, as this could affect their competition significantly?

A: This information is included in the Materials and methods part (lines 106-110)

The authors underline, that the results of this study could be of importance for tomato growers, but in which countries? Maybe only in the US or even a specific state. I suggest adding this information in the abstract.

A: We did not specify the country since the practical implication of this research could favor any grower regardless of their location. The tomato lines that were studied in this study may not be available in every country. Still, the research overall is a stimulus to invest and use weed-suppressive cultivars as a strategy for weed management, especially in a horticultural program for weed control.

Reviewer 3 Report

In this study entitled “Identification of weed-suppressive tomato cultivars for weed management” the authors examined the competitive ability of 17 tomato varieties against three important weeds. Weed management in tomato crop is critical in order to obtain high yield. Also, in several countries the number of registered herbicides for tomato crop is low. Thus, the implementation of integrated weed management programs is very important, while the cultivation of competitive cultivars can be an important part of these programs.

Thus, the topic of this study is very interest. The main disadvantage of this study is that presented very few experimental data, while the competitive ability of 17 tomato varieties was evaluated only in an early growth stage of tomato plants (28 days after sowing). One more disadvantage of this study, is that the authors they didn’t present data about growth parameters of tomato plants such as height, dry biomass of root system and aboveground-biomass. These data would help the authors to support their results. For the abovementioned reasons, this manuscript should be rejected.

Comments

Abstract: This section is well written.

Introduction: This section is well written. The author’s present useful information about competitive ability of tomato cultivars to weeds, while the negative impact of weeds in tomato crop was highlighted.

Line 38: “Wats.),t yellow” should be corrected to” Wats.), yellow”

Material and Methods: The authors should add information about the number of days from sowing to germination for tomato and three weed species. This is very important since the evaluation of competitive ability of tomato cultivars were made in earl growth stage of tomato (28 days after sowing).

In lines 298-299 the authors reported that “tomato plants could interfere with weed species either by competition and by generating chemical interference due to the release of allelochemicals from root exudates”. At 28 days after sowing, tomato plants have a small root system, so the release of allelochemicals should be also very low. For this reason, this part should be deleted. The authors should also explain why evaluate the completive ability of tomato only for this short period? One more measurement would show more about the competitiveness of the evaluated tomato varieties.

Figure 5: This figure should be deleted since is described in the text.  

Results: More data about growth parameters of tomato plants such as height, dry biomass of root system and aboveground-biomass should be added.

Discussion section: This section should be revised. The authors should add more information about the tomato traits that helps this species to exhibit high competitive ability.

In line 249 the authors reported that “the interference caused by the tomato cultivars on weed species varied widely”. This statement it is was supported in the text with data about growth of tomato plants.

Lines 250-257:  This part should be deleted since the authors they didn’t present information about the allelopathic ability of tomato. The authors should add the relevant references about allelopathic ability of tomato crop? Does the tomato plant show allelopathic ability? This question has not been supported either on the discussion or on the introduction section with relevant references.

Author Response

Abstract: This section is well written.

Introduction: This section is well written. The author’s present useful information about competitive ability of tomato cultivars to weeds, while the negative impact of weeds in tomato crop was highlighted.

Line 38: “Wats.),t yellow” should be corrected to” Wats.), yellow”

A: This change has been made.

Material and Methods: The authors should add information about the number of days from sowing to germination for tomato and three weed species. This is very important since the evaluation of competitive ability of tomato cultivars were made in earl growth stage of tomato (28 days after sowing).

A: This information was included in the materials and methods section (lines 106-110)

In lines 298-299 the authors reported that “tomato plants could interfere with weed species either by competition and by generating chemical interference due to the release of allelochemicals from root exudates”. At 28 days after sowing, tomato plants have a small root system, so the release of allelochemicals should be also very low. For this reason, this part should be deleted. The authors should also explain why evaluate the completive ability of tomato only for this short period? One more measurement would show more about the competitiveness of the evaluated tomato varieties.

A: Regardless of crop, the period of major weed interference is generally observed even at the early stages. At this time, most crops are ‘vulnerable’ to weed interference, especially crops that have a slow establishment and take a long time to cover the ground, leaving room for weed emergence. Based on our findings, there would not be any differences in plant height, biomass, or chlorophyll content after 28 days of planting. This is supported by other studies that show that the critical period for weed interference in tomato is between 24 to 35 days after emergence.

Figure 5: This figure should be deleted since is described in the text.  
A: This figure was added in the manuscript because one of the reviewers suggested including an illustration of the experimental setup even though it has been described in a sentence in the material and methods section.

Results: More data about growth parameters of tomato plants such as height, dry biomass of root system and aboveground-biomass should be added.

A: In this study, we only gathered data on the weed species as our primary focus was to identify suppression of the weed by tomato. Presenting growth parameters of tomato may not be helpful based on the goal of our study.  

Discussion section: This section should be revised. The authors should add more information about the tomato traits that helps this species to exhibit high competitive ability.

A: We have made significant changes in the discussion section of this paper.

In line 249 the authors reported that “the interference caused by the tomato cultivars on weed species varied widely”. This statement it is was supported in the text with data about growth of tomato plants. Lines 250-257:  This part should be deleted since the authors they didn’t present information about the allelopathic ability of tomato. The authors should add the relevant references about allelopathic ability of tomato crop? Does the tomato plant show allelopathic ability? This question has not been supported either on the discussion or on the introduction section with relevant references.

A: This sentence was deleted, and the discussion of the results was significantly modified.

Reviewer 4 Report

Dear Editor, Dear Authors,

thank you for considering me as a reviewer for the manuscript entitled: "Identification of weed-suppressive tomato cultivars for weed management".

The subject matter of the research presented in the manuscript is interesting to me and deserves to be explored.

Nowadays, non-chemical methods of fighting pathogens, insects and weeds are increasingly appreciated. Therefore, I believe that the research presented in this manuscript should be published. However, I believe that the manuscript still needs further improvement.

Below I list my comments:

1) The manuscript is poorly suited to the template requirements of the Plants journal.

2) Abstract - there is no short summary of the results of the research carried out here, along with showing their significance.

3) Keywords - should not repeat words appearing in the title of the work, and should also be arranged in alphabetical order.

4) I believe that the paragraph relating to the conclusions should be transformed into a separate chapter Conclusions.

5) In my opinion, chapters Introduction and Discussion are supported by too few publications.

I suggest the Authors consider my modest remarks, and above all, please expand the literature database on the subject matter under study. 

Author Response

Dear Editor, Dear Authors,

thank you for considering me as a reviewer for the manuscript entitled: "Identification of weed-suppressive tomato cultivars for weed management".

The subject matter of the research presented in the manuscript is interesting to me and deserves to be explored.

Nowadays, non-chemical methods of fighting pathogens, insects and weeds are increasingly appreciated. Therefore, I believe that the research presented in this manuscript should be published. However, I believe that the manuscript still needs further improvement.

Below I list my comments:

  • The manuscript is poorly suited to the template requirements of the Plants journal.
    • A: We have used the template provided for authors, but we went through the text again to make sure we followed the correct journal guidelines.
  • Abstract - there is no short summary of the results of the research carried out here, along with showing their significance.
    • A: This section was modified according to other reviewers’ suggestions.
  • Keywords - should not repeat words appearing in the title of the work and should also be arranged in alphabetical order.
    • A: The keywords have been modified.
  • I believe that the paragraph relating to the conclusions should be transformed into a separate chapter Conclusions.
    • A: A conclusion section is now included in the manuscript.
  • In my opinion, chapters Introduction and Discussion are supported by too few publications. I suggest the Authors consider my modest remarks, and above all, please expand the literature database on the subject matter under study.
    • A: We have made significant revisions in the discussion section of this paper. The literature supporting our results has been expanded.

Round 2

Reviewer 1 Report

Dear Authors,

The authors significantly improved the work, presented the purpose of the work much better, and also provided a research hypothesis, but without the null hypothesis. The full hypothesis should read as follows: "The hypothesis underlying tomato was that the tomato has the potential to suppress surrounding weeds through a higher competitive ability or allelopathy, ultimately leading to weed suppression, against the null hypothesis that the tomato has no potential to suppress weeds". This hypothesis may or may not be confirmed, but it should be verified by the Authors in the discussion of the results. The hypothesis should be put forward in the introduction to the work, right after the purpose of the work, and not in the methodology.

In the methodology, the authors added an unclear sentence "At the time of emergence, all tomato and weed seedlings were thinned to four per pot", what does this mean? explain exactly if 4 crops and 4 weed plants left? There cannot be any understatements here.

The discussion was supplemented and significantly expanded on the missing aspects of weed reaction to tomato plants, the behavior of chlorophyll in tomato and weed plants, which significantly contributed to the explanation of the existing dependencies. The authors proved, inter alia, that competition for light can be the main determinant of a reduction in tomato yield due to weed interference.

The conclusion is now summarizing and generalizing. The work deserves to be published after minor corrections.

Author Response

  • The hypothesis was included in the introduction section (lines 84-90) and was verified in the results (line 174).
  • We have modified the sentence regarding the number of plants per pot (lines 111-113).

Reviewer 3 Report

The authors improved both the discussion and material-methods sections. The main problem with this article is that the authors present information about weed suppressive ability of tomato cultivars without presented information about growth of tomato plants. In previous review report, I mentioned that data about growth parameters of tomato plants such as height, dry biomass of root system and aboveground-biomass should be added. The authors presented useful data about the weed suppressive ability of some tomato cultivars, but the authors did not support with data about tomato growth this suppressive ability. Unfortunately for this reason this manuscript should be rejected.

Author Response

We are thankful to the editor for understanding our research limitations, and for the helpful comments and suggestions.